# EQUIVALENT DISTANCE GEOMETRY ERROR FOR MOLECULAR CONFORMATION COMPARISON

## ABSTRACT

*Straight-forward* conformation generation models, which generate 3-D structures directly from input molecular graphs, play an important role in various molecular tasks with machine learning, such as 3D-QSAR and virtual screening in drug design. However, existing loss functions in these models either cost overmuch time or fail to guarantee the equivalence during optimization, which means treating different items unfairly, resulting in poor local geometry in generated conformation. So, we propose **E**quivalent **D**istance **G**eometry **E**rror (EDGE) to calculate the differential discrepancy between conformations where the essential factors of three kinds in conformation geometry (i.e. bond lengths, bond angles and dihedral angles) are equivalently optimized with certain weights. And in the improved version of our method, the optimization features minimizing linear transformations of atom-pair distances within 3-hop. Extensive experiments show that, compared with existing loss functions, EDGE performs effectively and efficiently in two tasks under the same backbones.

## 1 INTRODUCTION

Small organic molecules, conventionally a chemical or biomedical topic, are now being widely studied by artificial intelligence researchers. Based on prior natural science knowledge, machine learning models aid in multiple tasks on molecules including 3-D Quantitative Structure-Activity Relationship (QSAR) analysis, virtual screening for drug discovery, and protein-ligand binding analysis. Before any real industrial applications, molecular conformation generation draws particular concern. The task is also challenging as (1) conformation contains the 3-D geometry which can be translated or rotated, making it complicated to compare the generated one with ground-truth; (2) the number of molecules found in nature presents combination explosive growth, so the procedure of generating conformations is supposed to be fast enough for applicability.

In the past few years, different machine learning models have been proposed to handle the molecular conformation generation task. Generally, the molecules are featurized into token sequences (e.g. SMILES (Weininger, 1988)) or 2-D molecular graphs [1] (Gilmer et al., 2017) to feed the neural networks, while the way to obtain conformations is optional: (1) *Straight-forward* methods directly generate the 3-D coordinates of atoms from the inputs; (2) *Step-wise* methods first predict distances between atom pairs and then rebuild the conformations from the distances either via discrete algorithms (e.g. EDG (Havel, 1998)) or bi-level (Xu et al., 2021). As the expense of avoiding the troublesome comparison between conformations, *step-wise* methods usually adopt more complex architectures and endure relatively expensive time and space costs. Therefore, based on its innate advantage of high efficiency, *Straight-forward* methods require powerful assistance to perform equally great as *step-wise* ones. And a well-designed loss function can be helpful.

In recent studies, the simplest way to compare the conformations is to represent the conformations with atom position matrices $\boldsymbol{R}(\mathcal{G}) \in \mathbb{R}^{n \times 3}$ and calculate their Root-Mean-Squared Deviations (RMSD). Since conformation is invariant to spatial permutations (e.g. translation and rotation), a universal approach is to first find an optimal alignment. As an instance, Kabsch algorithm (Kabsch, 1976) provides a differentiable method to align two given conformations. We referred to the so-calculated RMSD as Kabsch-RMSD. However, using Kabsch-RMSD as the loss of neural networks

---

[1]The atoms and bonds inside the molecule form the vertexes and edges respectively.

is neither efficient nor effective: it needs to calculate the rotation matrix for every single confor-mation, which stifles batch processing and increases time cost; meanwhile, the rotation matrix is vulnerable to misplaced atoms and poorly-predicted regions (Mariani et al., 2013), so the RMSD between aligned and reference conformation cannot correctly reflect molecular similarity when in-put conformations are less likely.

To further improve the applicability of loss function on molecular conformation comparing, **Dis-tance Geometry** (DG) is proposed to encode a conformation into a vector of distances among the atoms, where the comparison is applied. The main idea of DG is to indirectly optimize bond lengths D, bond angles $\Phi$ and dihedral angles[2] $\Psi$ by optimizing the distances among 1-hop, 2-hop and more distant neighbors respectively in the molecular graph. However, when balancing time and effects, traditional distance geometry methods are faced with a dilemma: (1) If too many distant neighbors in the graph are considered, the loss calculation will be time-consuming. (2) If only close atom-pairs (e.g. 3-hop at most) are considered, the factors in the geometry will be treated unfairly. Based on the theoretical and experimental demonstration in Appendix B.1, the gradient allocated on bond angles $\Phi$ and dihedral angles $\Psi$ will become much smaller than that on bond lengths D when op-timizing atom-pair distances, resulting in contorted angles and poor local geometry. Therefore, the equivalence in optimizing different factors should be stressed in the comparison.

To push the boundary of efficient and effective loss function for molecular conformation generation, we propose **E**quivalent **D**istance **G**eometry **E**rror (**EDGE**). Firstly, EDGE exploits the essential factors inside the conformation geometry, including a series of bond lengths D, bond angles $\Phi$ and dihedral angles $\Psi$. Then the factors are jointly optimized with certain weights to guarantee that they are fairly treated. To avoid the high complexity of directly calculating the angles, we introduce Taylor Expansion to our method to approximate the angles with linear transformations of atom-pair distances within 3-hop. Extensive experiments prove that the improved version of EDGE performs as fast as optimizing 1/2/3-hop atom-pair distances and achieves better performance than existing popular loss functions on conformation comparison when applied to *straight-forward* backbones.

To conclude, our main contributions are threefold:

- We propose a novel loss function, EDGE, for *straight-forward* models on molecular con-formation generation, which optimizes the essential factors in the conformation geometry equivalently, realizing high consistency of generation.

- To avoid the high complexity and gradient explosion caused by direct calculation of angles, we introduce Taylor Expansion and Multiplier Truncation which achieves efficient model optimization with improved EDGE and guarantees its applicability.

- We conduct comprehensive experiments on several recently proposed benchmarks, in-cluding QM9(Wu et al., 2018), GEOM-QM9 and GEOM-Drugs(Axelrod & Gómez-Bombarelli, 2020). The outstanding performance on different metrics shows that EDGE is powerful on both definite conformation prediction and multiple conformation generation tasks, compared with existing loss functions. Besides, EDGE's efficiency is guaranteed via theoretical analysis and empirical test.

## 2 PROPOSED METHOD

### 2.1 PRELIMINARIES

**Notations** In the statements and equations below, we use italic letters for scalars and indices, bold lower-case letters for (column) vectors, bold upper-case letters for matrices, calligraphic letters for sets, and normal letters for annotations. Given a certain molecule, $\mathcal{M} = (\mathcal{V}, \mathcal{E}, n, m, X^{\mathrm{v}}, X^{\mathrm{e}})$ is the corresponding molecular graph. Here, $\mathcal{V}$ is the set of all $n$ atoms, $\mathcal{E} \subset \mathcal{V} \times \mathcal{V}$ is the set of all $m$ chemical bonds, $X^{\mathrm{v}} = (x_1^{\mathrm{v}}, ..., x_n^{\mathrm{v}})^{\top} \in \mathbb{R}^{n \times d_{\mathrm{v}}}$ is the matrix of atomic features, and $X^{\mathrm{e}} = (x_1^{\mathrm{e}}, ..., x_m^{\mathrm{e}})^{\top} \in \mathbb{R}^{m \times d_{\mathrm{e}}}$ that of bond features. $\mathcal{G}$ is the geometry of its conformation and $\boldsymbol{R}(\mathcal{G})$ is the matrix of atom positions. $\overline{d}(u, v; \mathcal{M})$ is the graph distance between atom $u, v$ in $\mathcal{M}$ and $d(u, v; \mathcal{G})$ is their Euclidean distance in $\mathcal{G}$. A conformation geometry $\mathcal{G}$ has three types of **factors** $\mathrm{D}(\mathcal{G}), \Phi(\mathcal{G}), \Psi(\mathcal{G})$, which refer to the set of bond length $d$, bond angle $\phi$ and dihedral angle $\psi$

---

[2]Dihedral angle is referred as torsion angle in some works.

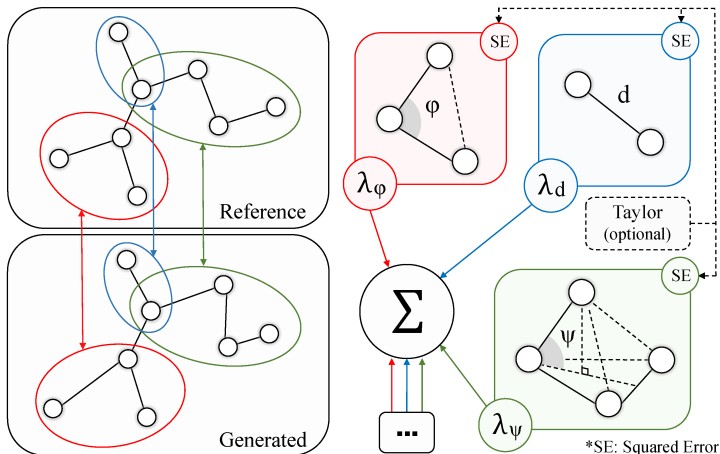

Figure 1: Given a generated conformation and its reference, different factors in the geometry $d, \phi, \psi$ are taken to calculate the Squared Errors, which are then added up with certain weights $\lambda_d, \lambda_\phi, \lambda_\psi$. As improvement, Taylor Expansion can be introduced as an optional approximation of SE, guaranteeing the high efficiency of EDGE. See 2.3 for details.

respectively. For operators, $\oplus$ is the operator of concatenations and $\| \cdot \|$ denotes the $l_2$ norm of the input vector.

**Problem Definition**    We consider following *straight-forward* molecular conformation generation process: the models input a molecular graph $\mathcal{M}$ and output a molecular conformation $\boldsymbol{R}(\mathcal{G})$. Our task is to find an efficient and effective loss function to calculate the differential discrepancy between generated and reference conformation $\mathcal{G}, \hat{\mathcal{G}}$, which can be utilized to optimize the models backwardly.

## 2.2    EQUIVALENCE IN MOLECULAR GEOMETRY

When applied to *straight-forward* backbones on molecular conformation generation, existing DG-based loss functions such as Conn-$k$ (Yang et al., 2021) can't fairly distribute the gradient to different factors $D(\mathcal{G})$, $\Phi(\mathcal{G})$ and $\Psi(\mathcal{G})$ in the conformation during optimization, resulting in contorted geometry. Simply calculating the average error between the generated $D(\mathcal{G}), \Phi(\mathcal{G}), \Psi(\mathcal{G})$ and the reference ones has the same deficiency, because the three types of factors are in different dimensions and equivalence can hardly be guaranteed while training the model. To fairly treat all the factors, it is necessary to jointly optimize them with certain weights which are determined by their corresponding degrees of freedom and value distributions in the dataset.

**Degree of Freedom**    Consider a 3-D molecular geometry $\mathcal{G}$ of a molecule with $n$ atoms, it has $f_{\mathcal{G}} = 3n - 6$ degree of freedom due to the existence of rotation and translation (Simm & Hernandez-Lobato, 2020). It means that a geometry can be just recovered from a certain combination of bond lengths, bond angles and dihedral angles with a total number $3n - 6$. Note that three types of factors occupy $f_D, f_\Phi, f_\Psi$ degrees of freedom out of $f_{\mathcal{G}}$ respectively, and we have[3]:

$$f_{\mathcal{G}} = f_D + f_\Phi + f_\Psi = 3n - 6 \qquad (1)$$

To completely perceive the local structure of molecular geometries, the number of bond lengths, bond angles and dihedral angles sampled in the geometry (i.e. $\hat{f}_D, \hat{f}_\Phi, \hat{f}_\Psi$) may exceed their exact degrees of freedom $f_D, f_\Phi, f_\Psi$ during optimization. So we need to balance them through a simple proportion (e.g. $f_D / \hat{f}_D$ for bond lengths $D$).

**Value Distribution**    When jointly optimizing bond lengths D, bond angles $\Phi$ and torsion angles $\Psi$, which are in different dimensions, it can hardly achieve equivalence to manually give hyper-parameters to balance the gradients. Therefore, we rescale $D, \Phi, \Psi$ with the reciprocals of their deviations $\sigma_D, \sigma_\Phi, \sigma_\Psi$ respectively to uniformly approximate a Gaussian distribution with $\sigma = 1$.

---

[3]The calculation of degrees of freedom for each kind of factors can be found in Appendix C.1.

### 2.3 Our Method: Equivalent Distance Geometry Error

We propose **E**quivalent **D**istance **G**eometry **E**rror (**EDGE**) as a loss function for *straight-forward* methods of molecular conformation generation. As illustrated in Fig.1, EDGE calculates the discrepancy between generated conformation $\mathcal{G}$ and reference conformation $\hat{\mathcal{G}}$ following[4]:

$$L(\mathcal{G}, \hat{\mathcal{G}}) = \lambda_{\mathrm{D}}^2 \sum_{\substack{d \in \mathrm{D}(\mathcal{G}) \\ \hat{d} \in \mathrm{D}(\hat{\mathcal{G}})}} (d - \hat{d})^2 + \lambda_{\Phi}^2 \sum_{\substack{\phi \in \Phi(\mathcal{G}) \\ \hat{\phi} \in \Phi(\hat{\mathcal{G}})}} (\phi - \hat{\phi})^2 + \lambda_{\Psi}^2 \sum_{\substack{\psi \in \Psi(\mathcal{G}) \\ \hat{\psi} \in \Psi(\hat{\mathcal{G}})}} (\psi - \hat{\psi})^2$$

$$\text{where} \quad \lambda_{\mathrm{D}} = \frac{f_{\mathrm{D}}}{\hat{f}_{\mathrm{D}}} \frac{1}{\sigma_{\mathrm{D}}} \quad \lambda_{\Phi} = \frac{f_{\Phi}}{\hat{f}_{\Phi}} \frac{1}{\sigma_{\Phi}} \quad \lambda_{\Psi} = \frac{f_{\Psi}}{\hat{f}_{\Psi}} \frac{1}{\sigma_{\Psi}}$$

(2)

To be more efficient, we can encode a molecular conformation into a vector $\boldsymbol{z}$:

$$\boldsymbol{z}_{\mathrm{D}} = [\lambda_{\mathrm{D}} d | d \in \mathrm{D}(\mathcal{G})] \quad \boldsymbol{z}_{\Phi} = [\lambda_{\Phi} \phi | \phi \in \Phi(\mathcal{G})] \quad \boldsymbol{z}_{\Psi} = [\lambda_{\Psi} \psi | \psi \in \Psi(\mathcal{G})]$$

$$\boldsymbol{z} = \boldsymbol{z}_{\mathrm{D}} \oplus \boldsymbol{z}_{\Phi} \oplus \boldsymbol{z}_{\Psi}$$

(3)

Then Eq.2 can be represented as the discrepancy between two encodings of conformation, which achieves fast comparison among multiple conformations:

$$L(\mathcal{G}, \hat{\mathcal{G}}) = \|\boldsymbol{z} - \hat{\boldsymbol{z}}\|^2$$

(4)

**Calculation of** D  For a conformation $\mathcal{G}$, we first calculate the distance matrix $\boldsymbol{D}$:

$$\boldsymbol{D}_{i,j} = \|\boldsymbol{R}(\mathcal{G})_i - \boldsymbol{R}(\mathcal{G})_j\|_2$$

(5)

where $\boldsymbol{R}(\mathcal{G})_i \in \mathbb{R}^3$ is the 3-D position of $i$-th atom in $\mathcal{G}$. Then the value of sampled bond lengths D can be found in $\boldsymbol{D}$.

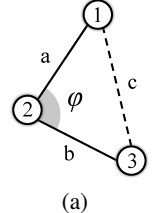

(a)

**Calculation of** $\Phi$  For a certain bond angle $\phi$ determined by three atoms in $\mathcal{G}$, $a, b$ are the lengths of its two sides and $c$ is the length of the side opposite it, which is shown in Fig.2(a). As $a, b, c$ can be found in $\boldsymbol{D}$, the value of $\phi$ can be calculated through the cosine theorem as

$$\phi = \arccos \frac{a^2 + b^2 - c^2}{2ab}$$

(6)

**Calculation of** $\Psi$  For a certain dihedral angle $\psi$ determined by four sequentially-connected atoms in $\mathcal{G}$, $a, b, c$ are the lengths of the sides in the atom chain and $d, e, f$ are 2/3-hop distances among the atoms, which is shown in Fig.2(b). Following Euler's formula in the tetrahedron, we have[5]:

$$\psi = \arcsin \sqrt{\frac{4a^2 b^2 e^2 - b^2 t_2^2 - a^2 r_2^2 - e^2 t_1^2 + r_2 t_1 t_2}{4a^2 b^2 c^2 - a^2 r_1^2}}$$

(7)

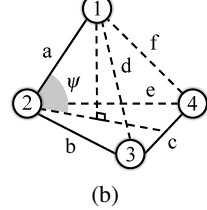

(b)

Figure 2: $\phi$ and $\psi$ in the conformation geometry.

### 2.4 Improvement

#### 2.4.1 Deficiency of Original EDGE

**High Complexity**  Although obtaining the exact value of $d, \hat{d}$ in $\mathrm{D}(\mathcal{G}), \mathrm{D}(\hat{\mathcal{G}})$ is straightforward, calculating and optimizing the factors $\Phi(\mathcal{G})$ and $\Psi(\mathcal{G})$ will bring high computational complexity due to the existence of complicated operations like inverse trigonometric functions.

**Gradient Explosion**  When $\phi \to \pi$ or $\psi \to 0, \pi/2$, the derivative of inverse trigonometric functions in Eq.6,7 reaches $\infty$. Even if we introduce Taylor Expansion to the calculation of each factor's error, which will be shown in 2.4.2, the same situation occurs when calculating the partial derivatives. So, gradient explosion is also a considerable problem during the optimization with EDGE.

---

[4]The calculation of the weights for different datasets can be found in Appendix C.

[5]In Eq. 7, $r_1 = b^2 + c^2 - e^2, r_2 = b^2 - c^2 + e^2, t_1 = a^2 + b^2 - d^2, t_2 = a^2 + e^2 - f^2$. And details of the calculation of $\psi$ are available in Appendix A.1.

### 2.4.2 TAYLOR EXPANSION

To solve the problem of high computational complexity and make calculation in Eq.2 efficient, we utilize Taylor Expansion to approximate the errors of bond angles $\Phi$ and dihedral angles $\Psi$ with a linear transformation of atom-pair distances in $\boldsymbol{D}$ from Eq.5.

For each factor-pair $\phi$ and $\hat{\phi}$ in $\Phi(\mathcal{G})$ and $\Phi(\hat{\mathcal{G}})$, considering the bond lengths $\hat{a}, \hat{b}, \hat{c}$ in $\hat{\mathcal{G}}$ which generate $\hat{\phi}$ in the way mentioned above, they have the corresponding bond lengths $a, b, c$ in $\mathcal{G}$. We can compose the vectors $\hat{\boldsymbol{u}} = (\hat{a}, \hat{b}, \hat{c})^\top$ and $\boldsymbol{u} = (a, b, c)^\top$ with them respectively. Like Eq. 6, $\phi$ can be taken as a function w.r.t $\boldsymbol{u}$. So, we can apply multivariate Taylor expansion to $\phi - \hat{\phi}$:

$$\phi - \hat{\phi} = (\frac{\partial \phi}{\partial \boldsymbol{u}}|_{\boldsymbol{u}=\hat{\boldsymbol{u}}})^\top (\boldsymbol{u} - \hat{\boldsymbol{u}})$$

$$\text{where} \quad \boldsymbol{u} - \hat{\boldsymbol{u}} \to \boldsymbol{0} \quad \text{and} \quad \frac{\partial \phi}{\partial \boldsymbol{u}} = (\frac{\partial \phi}{\partial a}, \frac{\partial \phi}{\partial b}, \frac{\partial \phi}{\partial c})^\top \tag{8}$$

Similarly, for each factor-pair $\psi$ and $\hat{\psi}$ in $\Psi(\mathcal{G})$ and $\Psi(\hat{\mathcal{G}})$, we can get the vectors $\hat{\boldsymbol{v}} = (\hat{a}, \hat{b}, \hat{c}, \hat{d}, \hat{e}, \hat{f})^\top$ and $\boldsymbol{v} = (a, b, c, d, e, f)^\top$. Like Eq. 7, $\psi$ can be taken as a function w.r.t $\boldsymbol{v}$. So we can apply multivariate Taylor expansion to $\psi - \hat{\psi}$:

$$\psi - \hat{\psi} = (\frac{\partial \psi}{\partial \boldsymbol{v}}|_{\boldsymbol{v}=\hat{\boldsymbol{v}}})^\top (\boldsymbol{v} - \hat{\boldsymbol{v}})$$

$$\text{where} \quad \boldsymbol{v} - \hat{\boldsymbol{v}} \to \boldsymbol{0} \quad \text{and} \quad \frac{\partial \psi}{\partial \boldsymbol{v}} = (\frac{\partial \psi}{\partial a}, \frac{\partial \psi}{\partial b}, \frac{\partial \psi}{\partial c}, \frac{\partial \psi}{\partial d}, \frac{\partial \psi}{\partial e}, \frac{\partial \psi}{\partial f})^\top \tag{9}$$

As the generated conformation $\mathcal{G}$ is changeable while reference conformation $\hat{\mathcal{G}}$ is fixed during training, the partial derivatives $\frac{\partial \phi}{\partial \boldsymbol{u}}|_{\boldsymbol{u}=\hat{\boldsymbol{u}}}, \frac{\partial \psi}{\partial \boldsymbol{v}}|_{\boldsymbol{v}=\hat{\boldsymbol{v}}}$, which is invariant to $\hat{\mathcal{G}}$, can be obtained in data processing[6]. Therefore, the discrepancy of bond angles and dihedral angles in Eq.2 can be replaced with Eq.8,9, which are simply linear transformations of atom-pair distances. Besides, the atom-pairs considered here is at most 3-hop connected, so the number of variables in linear transformation (i.e. atom-pair distances) is in the same order of magnitude with atom count $n$ (see Appendix B.2). So, the problem of high complexity of EDGE is resolved by Taylor Expansion.

### 2.4.3 MULTIPLIER TRUNCATION

Taylor Expansion solves the problem of high computational complexity, but gradient explosion still exists. Here, we apply **Multiplier Truncation** strategy to truncate the partial derivatives to finite values. Instead of limiting the final values, the problematic multipliers hidden in the formulas are found and then truncated to prevent gradient explosion, while the useful parts in the values are mostly preserved.

For $\phi$ w.r.t. $\boldsymbol{u}$, we have:

$$\frac{\partial \phi}{\partial \boldsymbol{u}} = \frac{\partial \phi}{\partial \cos \phi} \frac{\partial \cos \phi}{\partial \boldsymbol{u}} = -\frac{1}{\sin \phi} \frac{\partial \cos \phi}{\partial \boldsymbol{u}} \tag{10}$$

When $\phi \to \pi$, $\frac{1}{\sin \phi} \to \infty$. As stable values can always be acquired by multiplier $\frac{\partial \cos \phi}{\partial \boldsymbol{u}}$, $\frac{1}{\sin \phi}$ is the cause of gradient explosion in Eq.10. Here, we truncate $-\frac{1}{\sin \phi}$ to $[-10, -1]$ from $(-\infty, -1]$.

Similarly, for $\psi$ w.r.t. $\boldsymbol{v}$, we have:

$$\frac{\partial \psi}{\partial \boldsymbol{v}} = \frac{\partial \psi}{\partial \sin^2 \psi} \frac{\partial \sin^2 \psi}{\partial \boldsymbol{v}} = \frac{1}{2 \sin \psi \cos \psi} \frac{\partial \sin^2 \psi}{\partial \boldsymbol{v}} \tag{11}$$

When $\psi \to 0$ or $\pi/2$, $\frac{1}{2 \sin \psi \cos \psi} \to \infty$, so we truncate $\frac{1}{2 \sin \psi \cos \psi}$ to $[1, 10]$ from $[1, \infty)$. Moreover, inside the multiplier $\frac{\partial \sin^2 \psi}{\partial \boldsymbol{v}}$, the value of a denominator $4b^2 c^2 - (b^2 + c^2 - e^2)^2$ will tend to 0 when $e = b + c$ (i.e. the bond angle between $b, c$ is $\pi$), resulting in that the elements in the vector reaches $\infty$ and triggers explosion. To avoid this, we truncate $4b^2 c^2 - (b^2 + c^2 - e^2)^2$ to $[10, \infty)$ from $[0, \infty)$.

---

[6]Detailed formulas of the partial derivatives in Eq.8,9 are available in Appendix A.2.

The improved version of EDGE with Taylor Expansion and Multiplier Truncation resolves the previous deficiency and achieves time-friendly model training with equally powerful performance in the experiment.

## 3 EXPERIMENT

### 3.1 EXPERIMENT SETUP

#### 3.1.1 TASKS

We evaluate our loss function on two molecular conformation tasks: **definite conformation prediction** and **multiple conformation generation**.

**Definite conformation prediction** aims to predict the optimal conformation for each molecule. Here, we use a popular dataset **QM9** (Wu et al., 2018), which contains the conformations of about 133k molecules. These reference conformations are generated through Density Functional Theory (DFT), making sure that they are in a relatively stable status. The backbone model where we apply the loss functions is **PhysNet** (Yang et al., 2021), a learnable force field designed to optimize the molecular conformations. To show the performance of the loss functions on various aspects, they are evaluated on following metrics:

- D **RMSE**, $\Phi$ **RMSE** and $\Psi$ **RMSE**: the root-mean-squared error on the bond lengths, bond angles and dihedral angles sampled from molecular graphs, respectively.

- **A-RMSD** (Mansimov et al., 2019): the root-mean-squared derivation between reference conformation $\hat{\mathcal{G}}$ and generated conformation $\mathcal{G}$ after applying alignment function A[7], i.e.

$$\text{A-RMSD}(\mathcal{G}, \hat{\mathcal{G}}) = \text{RMSD}(\text{A}(\boldsymbol{R}(\mathcal{G}), \boldsymbol{R}(\hat{\mathcal{G}})), \boldsymbol{R}(\hat{\mathcal{G}})) \tag{12}$$

  where A is the alignment function.

- **lDDT-Score** (Mariani et al., 2013): Compute over all pairs of atoms in the reference structure at a distance closer than 15Å, and the distances in generated conformations are considered preserved if they are within a tolerance threshold compared to reference ones. lDDT-Score is the average fraction of preserved distances over thresholds 0.5Å, 1Å, 2Å and 4Å, i.e.

$$\text{lDDT-Score}(\mathcal{G}, \hat{\mathcal{G}}) = \frac{1}{4} \sum_{r \in \{0.5, 1, 2, 4\}} \underset{u, v \in \mathcal{V} \wedge d(u, v; \hat{\mathcal{G}}) < 15}{\text{average}} \mathbf{1}(|d(u, v; \mathcal{G}) - d(u, v; \hat{\mathcal{G}})| < r) \tag{13}$$

**Multiple conformation generation** aims to generate realistic and diverse conformations for a molecule. Here, we use the recent proposed **GEOM-QM9** and **GEOM-Drugs** (Axelrod & Gómez-Bombarelli, 2020), both of which contain multiple conformations for each molecule. Following the settings from Xu et al. (2021), we sampled 50,000 molecule-conformation pairs as training set and another 17813 pairs covering 150 molecules as test set from GEOM-QM9. As for GEOM-Drugs, we sampled 50,000 molecule-conformation pairs as training set and another 9161 pairs covering 100 molecules as test set. The backbone models selected to handle conformation generation are **PhysNet** and **CVGAE** (Mansimov et al., 2019), an auto-encoder designed for generating molecular conformations. Following metrics are used to evaluate the performance of loss functions:

- **COV-$\delta$** (Xu et al., 2021): the fraction of conformations in reference set $\mathbb{S}_r$ covered by at least one conformation in generated set $\mathbb{S}_g$ within a RMSD threshold $\delta$ after alignment, i.e.

$$\text{COV}(\mathbb{S}_g, \mathbb{S}_r) = \frac{1}{|\mathbb{S}_r|} \big| \{\hat{\mathcal{G}} \in \mathbb{S}_r | \text{A-RMSD}(\mathcal{G}, \hat{\mathcal{G}}) < \delta, \exists \mathcal{G} \in \mathbb{S}_g\} \big| \tag{14}$$

- **MAT** (Xu et al., 2021): the average RMSD over conformations in reference set $\mathbb{S}_r$, each of which is compared with the most similar conformation in generated set $\mathbb{S}_g$ after alignment, i.e.

$$\text{MAT}(\mathbb{S}_g, \mathbb{S}_r) = \frac{1}{|\mathbb{S}_r|} \sum_{\hat{\mathcal{G}} \in \mathbb{S}_r} \min_{\mathcal{G} \in \mathbb{S}_g} \text{A-RMSD}(\mathcal{G}, \hat{\mathcal{G}}) \tag{15}$$

---

[7]The alignment function is `AlignMol` from RDKit package v1.6.1.

- **Multi-lDDT-Score** (Mariani et al., 2013): the average lDDT-Score between each conformation in reference set $\mathbb{S}_r$ and the conformation in generated set $\mathbb{S}_g$ with highest lDDT-Score compared to the former, i.e.

$$\text{Multi-lDDT-Score}(\mathbb{S}_g, \mathbb{S}_r) = \frac{1}{|\mathbb{S}_r|} \sum_{\hat{\mathcal{G}} \in \mathbb{S}_r} \max_{\mathcal{G} \in \mathbb{S}_g} \text{lDDT-Score}(\mathcal{G}, \hat{\mathcal{G}}) \tag{16}$$

### 3.1.2 BASELINES

We compare EDGE with its variants and multiple loss functions for molecular conformation comparison as listed below:

- **Naive RMSD**: root-mean-squared error of the atom positions, i.e. $\text{RMSD}(\boldsymbol{R}(\mathcal{G}), \boldsymbol{R}(\hat{\mathcal{G}}))$
- **Kabsch RMSD** (Li et al., 2021): root-mean-squared error of the atom positions after aligning the conformations using Kabsch algorithm, i.e. $\text{RMSD}(\text{Kabsch}(\boldsymbol{R}(\mathcal{G}), \boldsymbol{R}(\hat{\mathcal{G}})), \boldsymbol{R}(\hat{\mathcal{G}}))$
- **lDDT-$\gamma$** (Mariani et al., 2013): root-mean-squared error of atom-pair distances whose values are less than $\gamma$ in reference conformation, i.e. $\underset{u,v \in \mathcal{V} \wedge d(u,v;\hat{\mathcal{G}}) < \gamma}{\text{RMSE}} (d(u,v;\mathcal{G}), d(u,v;\hat{\mathcal{G}}))$
- **Conn-$k$** (Yang et al., 2021): root-mean-squared error of distances of $k$-hop connected atom-pairs, i.e. $\underset{u,v \in \mathcal{V} \wedge \bar{d}(u,v;\hat{\mathcal{M}}) < k}{\text{RMSE}} (d(u,v;\mathcal{G}), d(u,v;\hat{\mathcal{G}}))$
- **EDGE-w/o $f$**: an invariant of EDGE, where degree of freedom is not considered when calculating hyper-parameters $\lambda_D, \lambda_\Phi, \lambda_\Psi$.
- **EDGE-w/o $\sigma$**: an invariant of EDGE, where value distribution is not considered when calculating hyper-parameters $\lambda_D, \lambda_\Phi, \lambda_\Psi$.
- **EDGE**: our loss function.

To further reflect how the loss function improves the performance, we also consider following baselines which are simply conformation generation strategies rather than learning models:

- *RDKit*: directly using the conformation generated by RDKit without further training.
- *Random Guess*: sampling the atom positions from the standard Gaussian distribution and constructing a conformation.

### 3.1.3 DETAILS OF IMPLEMENTATION

For PhysNet(RDKit initialized), we use RDKit (package version 1.6.1) to generate an initial conformation and then refine it with a PhysNet Block. The number and duration of time-step are 5 and 0.2 respectively. The hyper-parameter setup inside PhysNet follows Yang et al. (2021).

For CVGAE, we follow the settings of a UFF-free version from Mansimov et al. (2019), to show that EDGE can also perform better on generative backbones where derivative fine-tuning with force field is not considered.

The value of hyper-parameters $\lambda_D, \lambda_\phi, \lambda_\psi$ is calculated through sampling the degrees of freedom and standard deviation of $D, \phi, \psi$ from the train set of datasets QM9, GEOM-QM9 and GEOM-Drugs. The explicit process and the results of calculation can be found in Appendix C.2.

When optimizing the backbones, we set learning rate to `2e-6` for PhysNet and `5e-5` for CVGAE, and learning rate decay is `0.95`. We train all the models for 100 epochs on the train set.

### 3.2 RESULT

The result on three datasets and two backbones is listed in Tab.1, 2 and 3. More tables are presented in Appendix C.3.

As for efficiency, EDGE can optimize the molecular conformations as efficiently as Naive RMSD and Conn-3, and it's much faster than distance geometry methods considering distant atom-pairs

Table 1: Definite Conformation Prediction on QM9 with backbone PhysNet(RDKit initialized). **Time** refers to the seconds used to process around 106k molecules in a training epoch. ↑ means "higher is better".

| Baseline | Time | Global Geometry | | Local Geometry | | |
|---|---|---|---|---|---|---|
| | | A-RMSD | lDDT-Score(%)↑ | D RMSE | Φ RMSE | Ψ RMSE |
| *RDKit* | - | 0.750 | 83.3 | 0.084 | **0.088** | 0.354 |
| *Random Guess* | - | 1.420 | 38.3 | 0.709 | 0.825 | 0.526 |
| Naive RMSD | **439** | 1.151 | 42.0 | 0.452 | 0.702 | 0.476 |
| Kabsch RMSD | 3208 | 0.747 | 75.5 | 0.142 | 0.141 | 0.345 |
| Conn-3 | 478 | 0.736 | **85.4** | **0.038** | 0.093 | 0.326 |
| Conn-∞ | 1248 | 0.730 | 85.3 | 0.057 | 0.096 | 0.324 |
| lDDT-5 | 1105 | 0.733 | **85.4** | 0.043 | 0.096 | 0.325 |
| EDGE-w/o $f$ | 554 | 0.742 | 79.7 | 0.288 | 0.187 | 0.325 |
| EDGE-w/o $\sigma$ | 553 | 0.759 | 78.5 | 0.352 | 0.253 | 0.339 |
| EDGE | 554 | **0.710** | **85.4** | 0.069 | 0.097 | **0.318** |

Table 2: Multiple Conformation Generation on GEOM-Drugs with backbone PhysNet(RDKit initialized). **Time** refers to the seconds used to process 50k molecules in a training epoch. ↑ means "higher is better". An NaN result occurs in Naive RMSD.

| Baseline | Time | MAT | COV-0.5(%)↑ | COV-1.25(%)↑ | Multi-lDDT-Score(%)↑ |
|---|---|---|---|---|---|
| *RDKit* | - | 1.105 | 11.4 | 65.8 | 86.1 |
| *Random Guess* | - | 3.350 | 1.0 | 1.0 | 33.5 |
| Naive RMSD | **488** | - | - | - | - |
| Kabsch RMSD | 2412 | 1.172 | 9.9 | 62.2 | 76.6 |
| Conn-3 | 528 | 1.098 | 10.3 | 66.1 | 86.2 |
| EDGE | 579 | **1.076** | **11.9** | **68.6** | **86.3** |

(Conn-∞ and lDDT-5) and traditional RMSD with a time-consuming alignment algorithm (Kabsch RMSD). This is proved empirically in all three groups of experiments in the tables.

For **Definite Conformation Prediction** on QM9 (Tab.1), EDGE has the best performance on A-RMSD and lDDT-Score, two universal metrics to compare the conformations from a global view. To further demonstrate that EDGE outperforms the baselines through equivalently optimizing the bond lengths D, bond angles Φ and dihedral angles Ψ, we evaluate their RMSE on the three factors. Compared to traditional DG-based methods (Conn-$k$ and lDDT-$\gamma$), EDGE better retains the dihedral angles, which is significant in preserving complicated local structures like atom chains (see Sec.3.3). Moreover, EDGE's variants (w/o $f$ and w/o $\sigma$) break the balance on degrees of freedom or

Table 3: Multiple Conformation Generation on GEOM-QM9 with backbone CVGAE. **Time** refers to the seconds used to process 50k molecules in a training epoch. ↑ means "higher is better".

| Baseline | Time | MAT | COV-0.5(%)↑ | COV-1.25(%)↑ | Multi-lDDT-Score(%)↑ |
|---|---|---|---|---|---|
| *Random Guess* | - | 0.945 | 21.2 | 62.9 | 34.3 |
| Naive RMSD | **115** | 0.716 | 27.8 | 88.7 | 68.9 |
| Kabsch RMSD | 1254 | 0.670 | 28.7 | 95.8 | 72.6 |
| Conn-3 | 169 | 0.641 | 39.4 | 91.8 | 79.9 |
| EDGE | 180 | **0.582** | **41.0** | **99.7** | **81.6** |

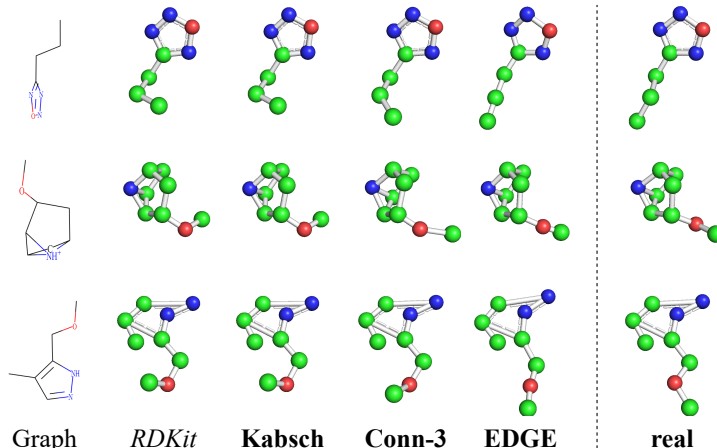

Figure 3: Results of visualization with PhysNet on QM9.

value distribution and lose the equivalence on D, $\Phi$, $\Psi$, so they obtain much worse results. In summary, EDGE treats D, $\Phi$, $\Psi$ equivalently, so it can avoid contorted local geometries and optimize the conformation more effectively.

The result of **Multiple Conformation Generation** shows that EDGE can also assist optimization in the generative conformation task and still has better performance compared to traditional Kabsch RMSD and DG-based Conn-3. Moreover, EDGE is adaptive to multiple types of backbones, no matter it's generative (CVGAE) or derivative (PhysNet).

In Tab.1,2 where the conformations are fine-tuned from RDKit-initialized ones, Kabsch RMSD and traditional DG-based methods may end up with results worse than *RDKit* in optimization. The main cause is that they can't measure the accurate discrepancy among the conformations, and the backbone falls into a solution which is optimal for the given loss function but suboptimal for universal metrics e.g. **A-RMSD**. Contrastively, EDGE finds out the essential factors in the conformation geometry and treats them equivalently when calculating the discrepancy, achieving a better performance.

### 3.3 VISUALIZATION

We visualize the conformations generated by PhysNet under multiple loss functions on dataset QM9 and compare them with RDKit-initialized conformations and reference ones, as it shows in Fig.3. Kabsch RMSD and Conn-3 retain appropriate bond lengths and bond angles while the atom chains inside are severely contorted because dihedral angles are significant in preserving local geometries, especially the atom chains in molecules (Winter et al., 2021). Contrastively, EDGE equivalently optimizes the essential factors in conformation geometry and focuses more on the angles. Therefore, PhysNet leveraged by EDGE produces conformations with better-structured atom chains as well as fine atom rings and achieves better performance.

### 4 CONCLUSION AND FUTURE WORK

We analyze the shortcomings of existing methods on measuring conformation discrepancy e.g. comparing the atom positions or atom-pair distances. Then we propose EDGE, which can find the essential factors in conformation geometry (i.e. bond lengths, bond angles and dihedral angles) and treat them equivalently by balancing the degrees of freedom as well as standard deviations. Theoretical analysis and extensive experiments demonstrate EDGE's efficiency and effectiveness when being applied to diverse backbone models. For future work, with the assistance of EDGE, the end-to-end conformation generation models are prevented from being misled by time-consuming or ineffective optimization strategies, and thus more well-designed models can be proposed and improved to handle various tasks on molecular conformation.

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

## A DETAILS OF EDGE

### A.1 THE FORMULA OF $\psi$

For a certain dihedral angle $\psi$ determined by four sequentially-connected atoms in $\mathcal{G}$ which is shown in Fig.2(b), we want to calculate the value of it using $a, b, c, d, e, f$, which are sides' lengths of the tetrahedron generated by the four atoms. Supposing that $\alpha$ is the angle between $b$ and $c$, the volume of the tetrahedron $V$ can be calculated following

$$V = \frac{1}{3}Sh = \frac{1}{3} \cdot \frac{1}{2}bc\sin\alpha \cdot a\sin\psi = \frac{abc\sin\alpha\sin\psi}{6} \tag{17}$$

Using Euler's formula in the tetrahedron, $V$ can also be calculated following

$$V = \frac{1}{6} \begin{vmatrix} b^2 & \frac{b^2+e^2-c^2}{2} & \frac{a^2+b^2-d^2}{2} \\ \frac{b^2+e^2-c^2}{2} & e^2 & \frac{a^2+e^2-f^2}{2} \\ \frac{a^2+b^2-d^2}{2} & \frac{a^2+e^2-f^2}{2} & a^2 \end{vmatrix}^{\frac{1}{2}} \tag{18}$$

Note that

$$\sin\alpha = \sqrt{1 - \cos^2\alpha} = \sqrt{1 - (\frac{b^2+c^2-e^2}{2bc})^2} \tag{19}$$

Combining Eq.17,18 and 19, we can get the formula of $\psi$:

$$\psi = \arcsin\sqrt{\frac{4a^2b^2e^2 - b^2t_2^2 - a^2r_2^2 - e^2t_1^2 + r_2t_1t_2}{4a^2b^2c^2 - a^2r_1^2}} \tag{20}$$

where

$$\begin{aligned} r_1 &= b^2 + c^2 - e^2 & r_2 &= b^2 - c^2 + e^2 \\ t_1 &= a^2 + b^2 - d^2 & t_2 &= a^2 + e^2 - f^2 \end{aligned} \tag{21}$$

### A.2 Partial Derivatives in Taylor Expansion

#### A.2.1 $\phi$

For each $\phi$ in $\Phi(\mathcal{G})$, considering the bond lengths $a, b, c$ in $\mathcal{G}$ which generate $\phi$ following 2(a), we can compose the vectors $\boldsymbol{u} = (a, b, c)^\top$ with them. Note that $\phi$ can be taken as a function w.r.t $\boldsymbol{u}$:

$$\phi = \phi(\boldsymbol{u}) = \arccos\frac{a^2 + b^2 - c^2}{2ab} \tag{22}$$

To implement Taylor Expansion, we need to calculate

$$\frac{\partial\phi}{\partial\boldsymbol{u}} = (\frac{\partial\phi}{\partial a}, \frac{\partial\phi}{\partial b}, \frac{\partial\phi}{\partial c})^\top \tag{23}$$

Following Eq.22, we can get the elements in Eq.23, which are the partial derivatives w.r.t. each variable:

$$\begin{aligned} \frac{\partial\phi}{\partial a} &= \frac{-a^2 + b^2 - c^2}{2a^2b\sin\phi} \\ \frac{\partial\phi}{\partial b} &= \frac{a^2 - b^2 - c^2}{2ab^2\sin\phi} \\ \frac{\partial\phi}{\partial c} &= \frac{c}{ab\sin\phi} \end{aligned} \tag{24}$$

#### A.2.2 $\psi$

For each $\psi$ in $\Psi(\mathcal{G})$, considering the bond lengths $a, b, c, d, e, f$ in $\mathcal{G}$ which generate $\phi$ following 2(b), we can compose the vectors $\boldsymbol{v} = (a, b, c, d, e, f)^\top$ with them. Note that $\psi$ can be also taken as a function $\psi(\boldsymbol{v})$ following Eq.20. To implement Taylor Expansion, we need to calculate

$$\frac{\partial\psi}{\partial\boldsymbol{v}} = (\frac{\partial\psi}{\partial a}, \frac{\partial\psi}{\partial b}, \frac{\partial\psi}{\partial c}, \frac{\partial\psi}{\partial d}, \frac{\partial\psi}{\partial e}, \frac{\partial\psi}{\partial f})^\top \tag{25}$$

Following $\psi(\boldsymbol{v})$, we can get the elements in Eq.25, which are the partial derivatives w.r.t. each variable:

$$\frac{\partial \psi}{\partial a} = \frac{a((4b^2c^2 - r_1^2)\sin^2\psi - s_1e^2 - s_2b^2 - s_3c^2 + 2a^2c^2)}{M}$$

$$\frac{\partial \psi}{\partial b} = \frac{b(2r_3a^2\sin^2\psi - s_1f^2 - s_2a^2 - s_3e^2 + 2b^2f^2)}{M}$$

$$\frac{\partial \psi}{\partial c} = \frac{c(2r_2a^2\sin^2\psi - 2r_2a^2 + t_1t_2)}{M}$$

$$\frac{\partial \psi}{\partial d} = \frac{d(-2t_1e^2 + r_2t_2)}{M} \tag{26}$$

$$\frac{\partial \psi}{\partial e} = \frac{e(2r_1a^2\sin^2\psi - s_1a^2 - s_2d^2 - s_3b^2 + 2d^2e^2)}{M}$$

$$\frac{\partial \psi}{\partial f} = \frac{f(-2t_2b^2 + r_2t_1)}{M}$$

where

$$r_1 = b^2 + c^2 - e^2$$
$$r_2 = b^2 - c^2 + e^2$$
$$r_3 = -b^2 + c^2 + e^2$$
$$s_1 = c^2 + d^2 - f^2$$
$$s_2 = c^2 - d^2 + f^2 \tag{27}$$
$$s_3 = -c^2 + d^2 + f^2$$
$$t_1 = a^2 + b^2 - d^2$$
$$t_2 = a^2 + e^2 - f^2$$
$$M = a^2(r_1^2 - 4b^2c^2)\sin\psi\cos\psi$$

# B PROVES

## B.1 PROVE OF INEQUITABLE GRADIENTS OVER D, Φ, Ψ IN DISTANCE GEOMETRY

We first analyze the gradients allocated on bond lengths D and bond angles Φ when optimizing 2-hop atom-pair distances. Suppose two bonds $a, b$ with a bond angle $\phi$, note that the third side of the triangle is $c$, a 2-hop distance (see Fig.2(a)). We have:

$$c^2 = a^2 + b^2 - 2ab\cos\phi \tag{28}$$

The partial derivatives of c on variables $a, b \in D$ and $\phi \in \Phi$ are:

$$\frac{\partial c}{\partial a} = \frac{a - b\cos\phi}{c} \qquad \frac{\partial c}{\partial b} = \frac{b - a\cos\phi}{c} \qquad \frac{\partial c}{\partial \phi} = \frac{ab\sin\phi}{c} \tag{29}$$

To further quantify the derivatives in Eq.29, we sample the distributions of D, Φ from QM9 dataset (see more details in Appendix C.2). We found that bond lengths has a mean value of $\mu_D \approx 1.452$ and standard deviation $\sigma_D \approx 0.103$, and bond angles has $\mu_\Phi \approx 108°$ and $\sigma_\Phi \approx 22°$. As $\sigma_D$ is relatively small compared to $\mu_D$, we suppose that $a = b = \mu_D$. Then the values of the derivatives are only variant to the temporary $\phi$, and we get:

$$\frac{\partial c}{\partial a} = \frac{\partial c}{\partial b} = \cos\frac{\pi - \phi}{2} \qquad \frac{\partial c}{\partial \phi} = \mu_D\sin\frac{\pi - \phi}{2} \tag{30}$$

Eq.30 is visualized in Fig.4. When optimizing the 2-hop distances, a large portion of gradient is allocated to fine-tuning the bond lengths D rather than the bond angles Φ, especially for bigger bond angles. Therefore, traditional loss functions based on distance geometry, where the bond angles are expected to be optimized by preserving 2-hop distances, didn't treat D, Φ equivalently and would cause contorted local geometries.

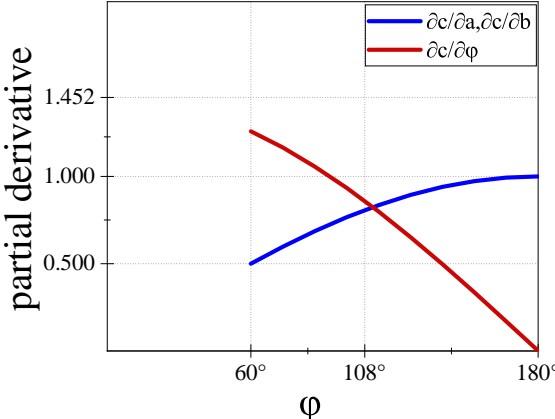

Figure 4: Visualization of partial derivatives in Eq.30. Ticks with $\mu_D = 1.452$ and $\mu_\Phi = 108°$ are highlighted.

When optimizing 3-hop atom-pair distances, the inequity of leveraging $D, \Phi, \Psi$ is proved empirically from the result in Tab.1, as the DG-based loss functions (Conn-$k$ and lDDT-$\gamma$) focus more on bond lengths and less on dihedral angles.

### B.2 PROVE OF LIMITED ATOM-PAIR NUMBER

For an atom in a common molecule, the number of neighbors is at most 6 (e.g. sulfur atom). Therefore, an atom has at most 30 2-hop neighbors and 150 3-hop neighbors, which means that at most 186 atoms are 3-hop connected to it. For a molecule with $n$ atoms, the number of 3-hop connected atom-pairs will not exceed $\frac{1}{2} \times 186n = 93n$.

Note that the number of 3-hop connected atom-pairs is $\gamma$ times of atom number $n$. For molecules in train set of **GEOM-Drugs** where the average atom number is around 44.2 (Xu et al., 2021), the estimated value of $\gamma$ is 4.6. This empirical result shows that optimizing the 1/2/3-hop atom-pair distances is constant times more expensive compared to directly optimizing atom positions, and it can be $n$ times faster than considering all atom-pairs, as the constant $\gamma = 4.6$ is much smaller than atom number $n = 44.2$.

## C IMPLEMENTATION DETAILS

### C.1 CALCULATION OF DEGREE OF FREEDOM

Consider a 3-D conformation geometry $\mathcal{G}$ and the corresponding molecular graph $\mathcal{M}$ (with $n$ atoms), the degrees of freedom is $f_\mathcal{G} = 3n - 6$ due to translation and rotation. We can figure out the exact number of degrees on three types of factors by:

- Generate a spanning tree $\mathcal{T}$ for the molecular graph $\mathcal{M}$, and $f_D = n - 1$ is the number of edges in $\mathcal{T}$.

- For every atom $v$ in $\mathcal{T}$, if the number of its neighbors $\mathcal{N}(v)$ are more than 1, it provides $2|\mathcal{N}(v)| - 3$ degrees of freedom for $f_\Phi$[8]. In brief, we have $f_\Phi = \sum_{v \in \mathcal{V}(\mathcal{T}) \wedge |\mathcal{N}(v)| > 1} 2|\mathcal{N}(v)| - 3$

- Obviously, $f_\Psi = f_\mathcal{G} - f_D - f_\Phi$

---

[8]For an atom $v$ whose distances to its neighbors $\mathcal{N}(v)$ are given, it needs $2|\mathcal{N}(v)| - 3$ exact values of bond angles ($v$ serves as the axis) to determine their relative position in 3-D space, if $|\mathcal{N}(v)| > 1$ (Havel, 1998).

Because there might be more than one possible spanning trees in a molecular graph, the output $f_D, f_\Phi, f_\Psi$ is diverse. However, no matter how $f_D, f_\Phi, f_\Psi$ compose the total degrees of freedom $f_\mathcal{G}$, it always embodies a set of $3n - 6$ factors (including bond lengths, bond angles and dihedral angles) which can determine the conformation geometry $\mathcal{G}$. Therefore, the correctness of output $f_D, f_\Phi, f_\Psi$ is insensitive to process of generating $\mathcal{T}$.

## C.2 Calculation of Hyper-parameters $\lambda$

To capture the complete local geometry, we consider all possible bond length, bond angles and dihedral angles when sampling them from conformation $\mathcal{G}$, and their numbers $\hat{f}_D, \hat{f}_\Phi, \hat{f}_\Psi$ may exceed the actual degrees of freedom $f_D, f_\Phi, f_\Psi$. So we balance them with weights $\frac{f_D}{\hat{f}_D}, \frac{f_\Phi}{\hat{f}_\Phi}, \frac{f_\Psi}{\hat{f}_\Psi}$. The weights are calculated through sampling from a certain dataset, whose exact values are listed in Tab.4.

Table 4: The proportion of degrees of freedom over number of sampled factors on $D, \Phi, \Psi$. To save time, we only sampled 1/10 molecules in datasets **GEOM-QM9** and **GEOM-Drugs**.

|  | QM9 | GEOM-QM9 | GEOM-Drugs |
|---|---|---|---|
| # of molecules | 133885 | 5000 | 5000 |
| Sum. of $f_D$ | 830932 | 38959 | 119701 |
| Sum. of $f_\Phi$ | 1030264 | 48274 | 153801 |
| Sum. of $f_\Psi$ | 311779 | 14644 | 70601 |
| Sum. of $\hat{f}_D$ | 1002205 | 46946 | 133582 |
| Sum. of $\hat{f}_\Phi$ | 1570148 | 75635 | 209399 |
| Sum. of $\hat{f}_\Psi$ | 1927138 | 94642 | 271308 |
| $f_D/\hat{f}_D$ | 0.829 | 0.830 | 0.896 |
| $f_\Phi/\hat{f}_\Phi$ | 0.656 | 0.638 | 0.734 |
| $f_\Psi/\hat{f}_\Psi$ | 0.162 | 0.155 | 0.260 |

We also sampled the standard deviation of $D, \Phi, \Psi$ for all three datasets, as it shows in Tab.5.

Table 5: The standard deviation of sampled $D, \Phi, \Psi$ in each dataset.

|  | QM9 | GEOM-QM9 | GEOM-Drugs |
|---|---|---|---|
| $\sigma_D$ | 0.103 | 0.101 | 0.109 |
| $\sigma_\Phi$ | 0.387 | 0.380 | 0.122 |
| $\sigma_\Psi$ | 0.433 | 0.437 | 0.394 |

Then we calculate the hyper-parameters via Eq.2, and the result is listed in Tab.6.

Table 6: The hyper-parameters weighting $D, \Phi, \Psi$ for each dataset.

|  | QM9 | GEOM-QM9 | GEOM-Drugs |
|---|---|---|---|
| $\lambda_D$ | 8.063 | 7.689 | 8.394 |
| $\lambda_\Phi$ | 1.692 | 1.676 | 6.257 |
| $\lambda_\Psi$ | 0.378 | 0.347 | 0.681 |

## C.3 More Results on Multiple Conformation Generation

See tables 7 and 8.

Table 7: Multiple Conformation Generation on GEOM-QM9 with backbone PhysNet. **Time** refers to the seconds used to process 50k molecules in a training epoch. ↑ means "higher is better".

| Loss | Time | MAT | COV-0.5(%)↑ | COV-1.25(%)↑ | Multi-lDDT-Score(%)↑ |
|---|---|---|---|---|---|
| *RDKit* | - | 0.399 | 77.2 | **93.3** | 64.5 |
| *Random Guess* | - | 0.945 | 21.2 | 62.9 | 34.3 |
| Naive RMSD | **203** | 0.993 | 28.7 | 45.8 | 47.1 |
| Kabsch RMSD | 1880 | 0.596 | 29.1 | **93.3** | 56.9 |
| Conn-3 | 232 | 0.458 | 69.1 | **93.3** | 62.1 |
| EDGE | 260 | **0.394** | **81.6** | **93.3** | **64.8** |

Table 8: Multiple Conformation Generation on GEOM-Drugs with backbone CVGAE. **Time** refers to the seconds used to process 50k molecules in a training epoch. ↑ means "higher is better".

| Loss | Time | MAT | COV-0.5(%)↑ | COV-1.25(%)↑ | Multi-lDDT-Score(%)↑ |
|---|---|---|---|---|---|
| *Random Guess* | - | 3.350 | 1.0 | 1.0 | 33.5 |
| Naive RMSD | **440** | 2.565 | **2.4** | 8.7 | 61.4 |
| Kabsch RMSD | 2201 | 2.520 | 1.7 | 8.2 | 61.6 |
| Conn-3 | 469 | 2.540 | 1.6 | 8.0 | 61.6 |
| EDGE | 498 | **2.492** | 1.7 | **8.8** | **62.0** |

