# OpenReview forum: "Equivalent Distance Geometry Error for Molecular Conformation Comparison"
_ICLR.cc/2022/Conference — ICLR 2022 Submitted_

### Official Review · Reviewer_AM3U · 2021-10-30

**Correctness:** 3
**Technical Novelty And Significance:** 2
**Empirical Novelty And Significance:** 3
**Recommendation:** 3
**Confidence:** 5

**Main Review:**

Strength:
- The paper is well-written and easy to follow. The method is clearly stated.
- The experimental part is sound. The effectiveness of the proposed method is well supported by the experiments on several benchmarks.

Weakness:
- The most interesting point for me is combining the new objective and PhysNet to design a new molecular graph generation model in the multiple conformation generation setting. However, there seem some technical issues regarding the generative model. For a generative model, it should be learned by matching the distribution instead of matching every two data points. It seems with the PhysNet backbone, you can generate different conformations with different initialization from RDKit, but how do you determine which reference confirmation should the generated one be compared to? If you just compare it with a uniformly sampled one from the dataset, it would not be theoretically sound for maximizing the likelihood of the dataset. Besides, the numbers in Tab.2 and Tab.3 show that CVGAE is much better than your proposed PhysNet framework. I think this is due to this technical issue.
- Some highly related reference is missed. You may do not need to compare with it in experiments, but the technical connection should be clarified. For example, in the abstract, you mentioned the bi-level method for learning 3D objectives. but a recent paper on this direction is missed in the reference. (Xu et al. "An End-to-End Framework for Molecular Conformation Generation via Bilevel Programming." ICML 2021.)
- Though improved over the RMSD-based method, however, there are many recent methods with more competitive results, such as the references you have cited "A generative model for molecular distance geometry" and "Learning neural generative dynamics for molecular conformation generation.". It will be better to also compare them because otherwise the proposed method will not be practically useful; or if the comparison is not meaningful the author should clearly clarify the reason in paper.

**Summary Of The Paper:**

The papers proposed a new objective function for comparing two 3D structures, in the molecular conformation context. Specifically, they proposed to compare the bond lengths, angles, and torsions, and design reweights to balance the three components. Furthermore, they improved the efficiency by approximating the comparison of angles and torsions with Taylor expansion. Experiments demonstrate that compared with directly comparing the aligned RMSD and other metrics, the proposed method is both more efficient and effective.

**Summary Of The Review:**

The paper proposed an interesting idea for molecular conformation comparison. However, comparison with some important baselines is missed, and there is some essential technical issue, and thus I vote for rejection. But I'm willing to change if I made any mistake or the author can address my concern.

---

### Official Review · Reviewer_fdL6 · 2021-11-02

**Correctness:** 3
**Technical Novelty And Significance:** 2
**Empirical Novelty And Significance:** Not applicable
**Recommendation:** 3
**Confidence:** 5

**Main Review:**

**Originality:** The idea of the work is novel, the area of *differentiable* advanced conformation generation loss function are quite promising and poorly explored.

**Clarity:** The author describes the motivation, goals and contribution of the work. The text, however, needs improvement (quality of graphs is not good, language needs spell-checking).

**Significance:** The novelty of the idea is high, but the significance is not proven. The experiment does not show a significant gap between the proposed loss and baselines.

**Drawbacks / questions**:
The claim of this work is that the new loss function is as accurate as best of its alternatives, and is faster to evaluate at the same time. The main accomplishment of the work is the comparison of different loss functions under various metrics. However, there are multiple issues that have to be addressed to improve the manuscript:

1. The proposed loss is dependent on multiple hyperparameters (ratios of degrees of freedom, dispersion, gradient clipping thresholds), which are derived from data or set manually by the authors. This makes questionable the claims on the accuracy of the loss, because the tests were done only on 2 datasets: QM9 and GEOM. More thorough testing should be done to justify the procedure.

2. The method is at most marginally better than the Conn-k loss, as is shown in the experimental section. The proposed loss is much more complex than Conn-k on the other hand and requires dataset-derived quantities. More arguments should be provided to show the advantage.

3. The effect of Taylor approximation on the accuracy of the loss is not studied. The authors provide too few details about this aspect of the method.

4. Plots are sloppy (see Fig. 4 which contains unnecessary whitespace). This creates a bad impression. Overall, the current form of the manuscript can not proceed until the mentioned issues are addressed.

Overall, the current form of the manuscript can not proceed until the mentioned issues are addressed.

**Summary Of The Paper:**

The work  introduces a custom loss function for the molecular conformation similarity. The authors combine this loss function with the PhysNet and GraphDG models to either predict a single conformation or to generate an ensemble of conformations. Other loss functions, including Naive RMSD, Kabsch algorithm, Conn-k and lDDT-k are compared under the same settings.

The main contributions of the paper are as follow: authors proposed a novel loss function for conformation generation, that can be used in *straight-forward* models, also provide Taylor Expansion and Multiplier Truncation techniques to speed up this metric.

**Summary Of The Review:**

The idea of the paper is novel, still experiments do not confirm practical significance of the method. The method is dependent on hyperparameters choice, still authors do not provide an advice how to find suitable ones.

---

### Official Review · Reviewer_AFf9 · 2021-11-02

**Correctness:** 4
**Technical Novelty And Significance:** 2
**Empirical Novelty And Significance:** 2
**Recommendation:** 5
**Confidence:** 4

**Main Review:**

Strengths
The approach is new and relatively effective. It sidesteps costly operations, including trigonometric functions, and applies some insightful approximations to improve speed and avoid exploding gradients. It is the fastest / best approach to including key dihedral information in the loss function of conformer generation approaches.

Weaknesses

While the Taylor series approach is nifty, the results are not unexpected and do not represent a large methodological innovation. Likewise, the experimental improvements are there, but the improvement in either times or accuracy w.r.t. conn-3 is just too subtle.



Questions

"it needs to calculate the rotation matrix for every single conformation, which stifles batch processing and increases time cost"  The rotation matrix can be found through a quaternion tensor operation. Is this not batchable?

Standard deviations or confidence intervals across multiple experiments could help confirm the magnitude of the improvements



**Summary Of The Paper:**

This paper proposes a new loss function for conformer generation tasks. These tasks essentially generated 3D point clouds from molecular graphs, and require a geometric similarity metric between generated samples and training data to drive the learning. The traditional metric of distance is RMSD based on aligned poses, but the authors suggest this is too time-consuming and instead propose a heuristic-weighted combination of bond distance between first other neighbors, and a Taylor-expansion of angles between second-order neighbors and dihedrals between 3rd order neighbors.

The metric is empirically found to perform well, based on both statistical metrics used in conformer generation tasks, by visual inspection of pathological cases, and because the timings are excellent

**Summary Of The Review:**

All in all, i find this paper informative and useful, but it does not provide enough truly novel insights for acceptance. I think it would make an outstanding workshop paper, or complement to some new conformer generative approach, but in and of itself it does not provide enough novelty and applied impact

---

### Official Review · Reviewer_vUYf · 2021-11-03

**Correctness:** 3
**Technical Novelty And Significance:** 3
**Empirical Novelty And Significance:** 3
**Recommendation:** 5
**Confidence:** 3

**Main Review:**

Novel & Significance:
1.	In Table 2, the paper lacks a comparison with previous methods like ConfGF, CGCF, VAE based methods and so on, which makes me hard to evaluate the effectiveness of the method.
2.	Although it is a new perspective to use D-$\Phi$-$\Psi$ for molecular conformation generation, I did not see a clear evidence about why it is helpful and how much improvement over previous is brought.

Clarification:
1.	In Eqn.(8), what does the $u$ stand for?

The authors should release their code for reproduction.


**Summary Of The Paper:**

The authors proposed a new method for molecular conformation generation. The authors choose to use bond lengths $d$, bond angles $\phi$ and dihedral angles $\psi$ for molecular conformation generation. The authors use Taylor expansion and multiplier truncation to improve models.  The authors conduct experiments on QM9 to verify the algorithm.


**Summary Of The Review:**

The paper lacks a clear motivation why this method, and many important baselines are missing.

---

### Decision · Program_Chairs · 2022-01-20

**Decision:**

Reject

**Comment:**

This paper proposes a new loss function for molecular conformation comparison to be used in generation tasks. All reviewers found the research topic is interesting, but the work lacks in multiple aspects. Major concerns include limited contributions and novelty, lack of comparison with prior methods, limited improvements, writing and clarity, etc. The authors did not provide any response during discussion. Given the consistency and extent of concerns, and lack of response, I recommend this paper be rejected at this time.